# Reference Genes across Nine Brain Areas of Wild Type and Prader-Willi Syndrome Mice: Assessing Differences in *Igfbp7*, *Pcsk1*, *Nhlh2* and *Nlgn3* Expression

**DOI:** 10.3390/ijms23158729

**Published:** 2022-08-05

**Authors:** Delf-Magnus Kummerfeld, Boris V. Skryabin, Juergen Brosius, Sergey Y. Vakhrushev, Timofey S. Rozhdestvensky

**Affiliations:** 1Medical Faculty, Core Facility Transgenic Animal and Genetic Engineering Models (TRAM), University of Muenster, Von-Esmarch-Str. 56, 48149 Muenster, Germany; 2Institute for Systems Genetics, West China Hospital, Sichuan University, Chengdu 610041, China; 3Copenhagen Center for Glycomics, Departments of Cellular and Molecular Medicine, Faculty of Health Sciences, University of Copenhagen, 2200 Copenhagen, Denmark

**Keywords:** Prader–Willi syndrome, SNORD116, Igfbp7, Pcsk1, Nhlh2, Nlgn3, Pcsk2, transcriptome, gene expression, brain, reference genes, brain regions, RT-qPCR, PWS-critical region, posttranscriptional regulation

## Abstract

Prader–Willi syndrome (PWS) is a complex neurodevelopmental disorder caused by the deletion or inactivation of paternally expressed imprinted genes at the chromosomal region 15q11–q13. The PWS-critical region (*PWScr*) harbors tandemly repeated non-protein coding IPW-A exons hosting the intronic *SNORD116* snoRNA gene array that is predominantly expressed in brain. Paternal deletion of *PWScr* is associated with key PWS symptoms in humans and growth retardation in mice (*PWScr* model). Dysregulation of the hypothalamic–pituitary axis (HPA) is thought to be causally involved in the PWS phenotype. Here we performed a comprehensive reverse transcription quantitative PCR (RT-qPCR) analysis across nine different brain regions of wild-type (WT) and *PWScr* mice to identify stably expressed reference genes. Four methods (Delta Ct, BestKeeper, Normfinder and Genorm) were applied to rank 11 selected reference gene candidates according to their expression stability. The resulting panel consists of the top three most stably expressed genes suitable for gene-expression profiling and comparative transcriptome analysis of WT and/or *PWScr* mouse brain regions. Using these reference genes, we revealed significant differences in the expression patterns of *Igfbp7*, *Nlgn3* and three HPA associated genes: *Pcsk1,* *Pcsk2* and *Nhlh2* across investigated brain regions of wild-type and *PWScr* mice. Our results raise a reasonable doubt on the involvement of the Snord116 in posttranscriptional regulation of *Nlgn3* and *Nhlh2* genes. We provide a valuable tool for expression analysis of specific genes across different areas of the mouse brain and for comparative investigation of *PWScr* mouse models to discover and verify different regulatory pathways affecting this complex disorder.

## 1. Introduction

### 1.1. Prader–Willi Syndrome

Prader–Willi syndrome (PWS) is a neurodevelopmental disorder with a complex multisystem involvement affecting one in 15,000 to 25,000 newborn children, and is caused by the abolished expression of paternally imprinted genes on chromosomal region 15q11–q13 [1]. The core symptoms associated with PWS can be traced back to the absence of the transcript from the minimal PWS-critical region (*PWScr*), which harbors several non-protein coding RNAs arranged in a tandemly repeated array, including IPW-A exons that host the intronically located C/D-box small nucleolar RNA 116 (SNORD116) [2,3,4,5]. The *PWScr* region has a homologue in mice located on chromosome 7qC. In both species, this region is similar in its structure and imprinting pattern, suggesting an equivalent biological function and qualifies genetically depleted mice as a suitable model system for the investigation of the pathogenesis of PWS [6]. Notably, the expression of the *PWScr* derived non-protein coding RNAs is highly abundant in the brains of both species, but in humans, it is also detected in other tissues [2,7].

PWS is a multisystemic disorder with a characteristically biphasic symptomatology: starting from the reduced body weight of the embryo, neonatal hypotonia, including feeding difficulties and general failure to thrive, followed by a lack of satiety and hyperphagia, resulting in extreme weight gain, leading to morbid obesity [8]. An underlying problem in PWS patients starting from an early age is a dysregulated endocrine axis, including a deficiency in growth hormone (GH) [9,10]. Since the molecular mechanisms that give rise to growth hormone deficiency are poorly understood, researchers investigated other important components of the hypothalamic–pituitary pathway. Relevant genes which are affected in both human PWS patients, PWS-patient-induced pluripotent stem cell (iPSC) derived neurons and PWS mouse models include insulin-like growth factor-binding protein 7 (Igfbp7), proprotein convertase subtilisin/kexin type 1 (Pcsk1) and nascent helix-loop-helix 2 (Nhlh2) [11,12,13]. *Pcsk1* encodes a protease designated as prohormone convertase 1, whose proteolytic activation of prohormones, such as pro-GH-releasing hormone, increases their respective bioactivity [14,15]. The product of *Nhlh2* is a transcription factor which binds within the promoter region of *Pcsk1*, enhancing its transcription [16]. Both *Nhlh2*-null and *Pcsk1*-hypomorphic mouse models recapitulate core aspects of the PWS symptomatology, such as an impaired hypothalamic axis, growth retardation and eventually obesity [17,18,19,20]. Yet, a plausible mechanistic connection between the non-protein coding RNAs of the *PWScr* and the hypothalamic–pituitary axis remains elusive. The connection of PWS and *Igfbp7* is also not fully understood. Since children with PWS are deficient in GH and at the same time hypersensitive to GH as indicated by their increased levels of serum insulin-like growth factor-I (IGF-I) after GH treatment, compared to non-PWS patients deficient in GH, the regulation of IGF-I by IGF binding proteins may be disturbed in PWS [9,21].

In an attempt to elucidate the connection of the non-protein coding RNAs of the PWS locus with potential physiological pathways in a murine model system, one has to consider that *PWScr* transcripts are, unlike in humans, *de facto* restricted to the brains of mice. One suitable approach could therefore be to further resolve transcriptome changes within the brain and investigate whether the observed differences are limited to, for example, the hypothalamic region, or if they have consequences in other regions of the central nervous system (CNS) as well. To the best of our knowledge, no differential expression analysis of genes involved in the relevant pathways has been done until now.

### 1.2. Reference Genes for Expression Analysis across Brain Regions

The mammalian brain is composed of several anatomical structures with distinct functions. Differences between these brain compartments arise from specific cellular populations and are related to the respective regional transcriptome [22]. Gene expression analysis, especially in specific sub-compartments of the brain, should therefore reflect the intrinsic heterogeneity of the tissue.

The most meaningful transcriptomic measurements rely on relative quantification, that is, the measurement of the expression ratio of target genes to suitable reference genes [23,24,25]. This can, to a certain degree, account for variabilities in sample quality thereby allowing for a higher reproducibility and comparability of experiments [26,27]. Despite the recent cost reduction in high-throughput techniques, such as (single cell) RNA-sequencing and microarrays, these methods are still routinely supported by reverse transcription quantitative real-time PCR (RT-qPCR) for the validation of results and the normalization of data [28]. RT-qPCR is a robust, fast and comparatively low-priced method for the accurate quantification of gene expression. Its validity relies, however, on the availability of suitable housekeeping genes with an ideally constant expression level across all biological samples within one experiment [29,30]. The choice of appropriate reference genes is therefore a critical step in any experiment using data obtained by qPCR and should be treated with due diligence. Historically, basal metabolic genes, such as glyceraldehyde 3-phosphate dehydrogenase (*GAPDH*), cytoskeletal genes, such as β-actin (*ACTB*), and the ribosomal RNA genes (*18S and 28S rRNA*) have been widely used as reference genes in vertebrate samples [31,32] due to their high and relatively stable expression level over all cell types and tissues. However, even at the onset of the qPCR era, it was realized that these genes were often poor choices for the normalization of expression data since their mRNA levels vary by as much as 15-fold between tissues (i.e., *GAPDH*), are highly influenced by common cell culture components (i.e., *ACTB*) and also could be affected by any dysregulation of RNA polymerases I (i.e., *rRNAs*) and II, as is the case in numerous cancers [33,34,35,36].

Since the selection of reference genes is such a crucial step, a number of mathematical models to objectively identify the most suitable candidates within the investigated set of genes were developed. The four most commonly used methods and programs are the Delta Ct method [37], geNorm [38], BestKeeper [39] and NormFinder [40]. Using these tools, researchers were able to select the most suitable reference genes and compile specific panels of reference genes for the normalization of expression data adapted to their respective experimental design. Resulting reference genes cover a wide spectrum of research topics, such as human brain samples from patients with neurodegenerative disease [41], mouse CNS tissue during development and injury [42], rat brain regions after induced seizures [43] and specific tissue regions, such as mouse choroid plexus [44].

In this study, we performed comprehensive expression profiling of a selected set of 11 putative housekeeping genes in 9 different brain regions of WT and *PWScr* mice, respectively. We identified the three most stably expressed genes and combined them into a reference gene panel for the broad analysis of differentially expressed genes in nine investigated brain regions of WT and *PWScr* mice. In addition, our reference gene panel allows a comparative examination of gene expression between both genotypes in all nine brain regions. Subsequently, we analyzed the differential expression of *Igfbp7, Nlgn3*, *Pcsk1*, *Pcsk2*, and *Nhlh2* genes in different brain regions of WT and *PWScr* mice. Dysregulation of these genes in PWS model systems has been reported previously [13,45]. Moreover, due to the base-complementarity of SNORD116 to NHLH2 and NLGN3 mRNAs regions, the snoRNA was predicted to be involved in the posttranscriptional regulation of these genes. Our expression data demonstrate that the studied genes exhibit differential expression patterns in the analyzed brain regions of WT and *PWScr* mice. Although *PWScr^m+/p−^* mice showed an increased level of *Igfbp7* and a downregulation of *Pcsk1*, we could not confirm a decrease in *Pcsk2* expression and observed an increase in Nhlh2 mRNA abundance. Similarly, while our results revealed a trend of increased Nlgn3 mRNA levels in almost all brain regions of *PWScr^m+/p−^* mice, this effect was not specific for the particular predicted Snord116 mRNA-target, but for all spliced RNA isoforms examined.

## 2. Results

### 2.1. Partitioning the Mouse Brain into Nine Anatomical District Regions and Reference Gene Candidate Selection

Since the generation of the first PWS animal models to study the disorder, to the best of our knowledge, no comprehensive analysis has been performed to identify and evaluate stably expressed reference genes across different brain regions between wild-type and PWS-like genotypes. Moreover, we could not find any experimentally validated reference genes for the accurate normalization of expression data obtained from these nine brain areas of wild-type mouse in the literature (Figure 1A). Therefore, as a precondition for our studies, we selected and experimentally verified a panel of stably expressed genes across different anatomical regions of the mouse brain (Figure 1A). Initially, we selected 11 reference gene candidates: *Sdha*, *B2M*, *β-Actin*, *Alg5*, *Mogs*, *Hmbs*, *Gusβ*, *Man2b2*, *Cyc1*, *Snhg12* and *Tfrc*. Although a number of studies have reported the low expression stability of *Sdha*, *B2M* and *β-Actin* genes between different cell lines, tissues, and animal models of human diseases, we included them here due to their prevalence as reference genes to quantify RT-qPCR data.

For the remaining genes, we based our choices on RNA-seq data and other RT-qPCR reports detecting their stable expression either between several brain regions, in the developing brain, or in studies of various neurodegenerative diseases.

Among them are *Alg5* (Asparagine-linked glycosylation 5 homolog), *Man2b2* (Mannosidase alpha class 2B member 2) and *Mogs* (Mannosyl-oligosaccharide glucosidase), that are protein-coding genes involved in glycosylation. By analyzing the RNA-seq data reported recently, we were able to identify them as stably expressed genes in the mouse isocortex and cerebellum [46].

The *Hmbs* (hydoxymethylbilane sythase) gene encodes an enzyme that catalyzes a step in the formation of the porphyrin ring during heme biosynthesis. *Gusβ* (glucuronidase beta) is a lysosomal hydrolase enzyme, which cleaves glycosaminoglycans. Both *Hmbs* and *Gusβ* were identified as suitable reference genes during the investigation of developing mouse CNS tissue [42].

*Cyc1* (Cytochrome C1) encodes a subunit of the mitochondrial respiratory chain. It is conserved among a wide variety of species, including animals, yeast and plants. *Cyc1* is commonly used as a RT-qPCR reference gene in the central nervous system and was among the most stably expressed genes during the study of several neurodegenerative diseases (Alzheimer’s disease, Parkinson’s disease, multiple system atrophy, and progressive supranuclear palsy) in human prefrontal cortex and cerebellum tissue samples [41].

*Snhg12* (small nucleolar host gene 12) is a long non-protein coding transcript hosting three small nucleolar H/ACA box RNAs (Snora16a, Snora44, Snora61) and one C/D box snoRNA (Snord99), which have been evaluated as potential reference gene candidates before [47]. Although the expression of these individual small snoRNAs were not found to be particularly stable across 20 different tissues in humans, we wished to evaluate the expression stability of their long non-protein coding host transcript in different regions of central nervous system tissue. 

*Tfrc* (transferrin receptor) is a gene encoding an essential part of cellular iron recognition and subsequent endocytosis. It was ranked as the top reference gene in rat granule neurons during an investigation of multiple sclerosis [48].

After an initial screening process as determined by RefFinder comprehensive ranking [49] (see below), we identified the top eight stably expressed genes in nine investigated brain areas of wild-type (WT) mice. As expected, due to low stability in expression during preliminary screening, we removed *Sdha*, *B2M* and *β-Actin* from the reference gene candidate pool and completed the study with the eight remaining candidates: *Alg5*, *Mogs*, *Hmbs*, *Gusβ*, *Man2b2*, *Cyc1*, *Snhg12* and *Tfrc,* respectively (Figure 1B).

### 2.2. Compilation of a Reference Gene Panel

In order to identify the most suitable (meaning stably expressed) set of genes for WT and *PWScr^m+/p−^* mice, we ranked all eight candidates with four different methods: geNorm [38], Normfinder [40], BestKeeper [39] and comparative delta Ct [37]. We first assessed the expression stability of selected reference genes within all investigated brain areas of WT and *PWScr^m+/p−^* genotypes separately (Figure 2A–D). The ranking methods applied resulted in a rank order, placing the reference gene candidates in a sequence from most to the least stably expressed (Figure 2A–D). Subsequently, the average stability ranking was determined by calculating the geometric mean of the four ranking positions for each gene and arranging the genes accordingly from the most stable (i.e., lowest average rank) to the least stable (i.e., highest average rank). These four methods are combined into a single web-based tool, which calculates all investigated genes based on their expression stability, resulting in a final ranking comprised of the corresponding geometric mean of the resulting RT-qPCR values [49].

When nine different brains areas of WT mice were analyzed, the three most stably expressed genes across all analyzed regions were selected: *Alg5*, *Gusβ* and *Mogs,* respectively (Figure 2A,B). The ranking results among all four methods were mostly congruent with all algorithms placing the best three genes by average stability ranking within the four most stable genes (Figure 2A). For the brain regions of *PWScr^m+/p−^* mice, the most stably expressed genes determined by average stability ranking were *Alg5*, *Mogs* and *Hmbs*, respectively (Figure 2C,D). These results were also consistent, with the exception of BestKeeper, which ranked *Hmbs* only in the sixth position (Figure 2C). At the other end of the expression stability spectrum, we also found that all ranking methods produced comparable results, with *Tfrc* and *Snhg12* genes being classified as the least stably expressed among all nine brain regions. The only exception again was BestKeeper in the *PWScr^m+/p−^* mice, which ranked *Snhg12* at position 3 (Figure 2C).

Next, we performed a comparative ranking of the expression stability for each of the investigated brain areas between WT and *PWScr^m+/p−^* siblings (Appendix A). The results obtained for average expression stability ranking are represented in Figure 2F.

Finally, the reference gene panel was compiled by calculating the average expression stability rank from the inter-regional stability ranking (weighted at 50%) and the intra-regional ranking (weighted at 50%). The three genes with the best final ranking (*Alg5*-*Hmbs*-*Gusβ*) were henceforth used as housekeeping reference genes for our RT-qPCR analyses (Figure 2G, bottom row). 

It should be noted that the expression stability of certain genes for each of the studied brain regions between WT and *PWScr^m+/p−^* mice varies somewhat between the two genotypes. For example, we found *Alg5* to be the most stably expressed gene across most brain regions and the genotypes. However, it differs between midbrain samples of WT and *PWScr^m+/p−^* mice and ranks lower than the other five gene candidates (Figure 2F). Similarly, *Gusβ*, which is stably expressed in most of the brain areas, was actually the least stably expressed gene when comparing the olfactory bulb between WT and *PWScr^m+/p−^* animals (Figure 2F). Thus, if only a specific brain region is of interest, it might be more suitable to use a different combination of reference genes for analysis than our final housekeeping panel (Appendix A).

### 2.3. Effect of Snord116 on the Regional Expression of Igfbp7

Growth hormone deficiency is common among PWS patients, which in part accounts for impaired growth rate, short stature, muscle hypotonia, abnormal body composition, and low energy expenditure, which are well-known symptoms of this disorder. Interestingly, it was recently reported that *IGFBP7* was expressed at higher levels in PWS patients when compared to a healthy control population but decreased after growth hormone treatment [13]. Similar results were obtained when the expression level of *Igfbp7* was evaluated in the total brain of WT and *PWScr^m+/p−^* mice at postnatal day 7 (P7) [13].

Snord116 might influence *Igfbp7* gene expression via the regulation of proconvertases PCSK1 and/or PCSK2, which were found to be downregulated in iPSC-derived neurons from PWS patients [13].

Using our reference gene panel, we performed a comparative analysis of *Igfbp7*, *Pcsk1,* and *Pcsk2* expression in nine brain regions of *PWScr^m+/p−^* and WT mice (Figure 3A–C).

Overall, the relative *Igfbp7* gene expression was high in all investigated brain regions of both genotypes, with up to 2,5-fold differences between various areas within each genotype (Figure 3A). In agreement with previous results, we detected a tendency toward increased *Igfbp7* gene expression throughout all nine brain regions of *PWScr^m+/p−^* mice, with statistically significant upregulation in the hippocampus, hypothalamus, and cerebellum (Figure 3A). Although the observed differences between some brain regions of WT and *PWScr^m+/p−^* mice were not statistically significant, increasing the number of biological replicates, as in the case of the WT hypothalamus, could render them statistically significant.

When *Pcsk1* gene expression was examined, we detected small but statistically significant downregulation in the hypothalamus and pons regions of the brain of *PWScr^m+/p−^* mice compared to WT animals, while no changes in *Pcsk1* gene expression were detected in other regions (Figure 3B). In both *PWScr^m+/p−^* and WT mice, measured *Pcsk1* levels were highest in hypothalamus, olfactory bulb and isocortex and lowest in hippocampus; however, the putative functional significance of the observed mRNA-level changes must be further demonstrated at the protein level.

Surprisingly, and contrary to other reports of downregulated *PCSK2* expression in iPSC derived neurons from PWS patients [12], we found *Pcsk2* to be somewhat upregulated in four brain regions of *PWScr^m+/p−^* mice (Figure 3C). Regions with a statistically significant increase of Pcsk2 transcript were the olfactory bulb, isocortex, hippocampus and medulla. The expression of *Pcsk2* in other brain areas was similar between both *PWScr^m+/p−^* and WT mice. In both genotypes, *Pcsk2* was highly expressed in isocortex and to a lesser extent in hippocampus and thalamus, while the lowest expression was detected in olfactory bulb, cerebellum and pons of the brain (Figure 3C).

Recent cell culture experiments using iPSC-derived neurons from PWS patients and the mouse hypothalamic N29/2 cell line identified NHLH2 mRNA as a potential target for SNORD116 RNA [12,50]. The snoRNA contains a sequence that is complementary to NHLH2 mRNA, and it was suggested that this interaction could increase RNA stability [50]. In contrast to the human iPSC results [50], we detected increased levels of Nhlh2 mRNA in several brain regions of *PWScr^m+/p−^* mice compared to the WT animals, with a pronounced and statistically significant upregulation in the cerebellum, midbrain and hippocampus (Figure 3D). The cerebellum also had the highest levels of Nhlh2 mRNA in both genotypes, followed by thalamus and hypothalamus. Nhlh2 mRNA levels were extremely low in olfactory bulb and almost undetectable in the isocortex of analyzed animals. In contrast to *Nhlh2* gene expression, *Snord116* has approximately the same expression level in all analyzed brain areas of WT mice, questioning a direct involvement of the snoRNA in *Nhlh2* mRNA posttranscriptional regulation (Figure 3E).

### 2.4. Lack of Snord116 Influences Abundance but Not Splicing of Nlgn3 mRNA In Vivo

Phylogenetic analysis of the conserved Snord116 RNA guide sequence (called ASE—antisense element) revealed sequence complementarity to the exonic regions of several mRNAs. Among the possible candidates is alternatively spliced exon 3 of Neurolignin 3 (Nlgn3) pre-mRNA (Figure 4A). The Snord116 targeting region on Nlgn3 is evolutionary conserved in mammals, including humans and mice, and located in close proximity to the intron 2–exon 3 junction, overlapping a predicted exon splice enhancer. The hybridization energy of the Snord116–Nlgn3 base–pair interaction is about −20 kcal/mol, which is similar to that of regular C/D Box snoRNA-rRNA targeting [45]. In humans, the *NLGN3* gene undergoes posttranscriptional regulation during alternative splicing resulting in several different mRNA isoforms, at least two of which contain exon 3 [51]. Murine Nlgn3 pre-mRNA is processed in a similar manner. The UCSC Genome Browser on Mouse (GRCm39/mm39) annotates three potential splice isoforms and among them, only one contains exon 3. Interestingly, the NLGN3 splice isoform-dependent regulation of synaptic transmission in the hippocampus has been shown [52]. Recently, it was reported that Snord116 could regulate alternative splicing and increase the abundance of NLGN3 isoforms that contain exon three [45]. The experiments were performed in a human HeLa S3 cell line-based Snord116 knock-down system, where the Snord116 level was reduced by about 50%, using specific antisense oligonucleotides [45].

To confirm this finding in vivo, we designed three TaqMan assays to measure the expression levels of different Nlgn3 mRNA isoforms in nine brain regions of *PWScr^m+/p−^* and WT mice (Figure 4A,C). The first assay was designed on exons 5 and 6 to detect all putative splice isoforms. (Figure 4A,B). The second assay was specific for an isoform that includes exon 3 (Nlgn3 isoform 1, Figure 4A,C) and the third was designed for isoforms lacking exon 3 (Nlgn3 isoform 2, Figure 4A,D). In accordance with the previous results in cell culture [45], elimination of Snord116 cluster in *PWScr^m+/p−^* mice resulted in a slight (10–15%) increase of total Nlgn3 mRNA level in all brain regions except pons, although the difference only reached statistical significance in the hypothalamus (Figure 4B). Regardless of genotype, *Nlgn3* was highly expressed in all brain regions, with the highest mRNA level being detected in the hippocampus, isocortex, olfactory bulb, and hypothalamus and the lowest in the cerebellum and medulla. There were no changes in the regional expression profiles of *Nlgn3* isoforms between *PWScr^m+/p−^* and WT mice (Figure 4B).

We could also confirm an upregulation of Nlgn3 isoform 1 in *PWScr^m+/p−^* mice. The results were statistically significant for all nine brain regions, including pons (Figure 4C). The highest levels were detected in the thalamus, hippocampus, isocortex, and olfactory bulb. Interestingly, the lowest levels of Nlgn3 isoform 1 were detected in the hypothalamus for both genotypes.

However, in contrast to the cell culture experiments, we could not detect the downregulation of Nlgn3 mRNA isoforms 2 in *PWScr^m+/p−^* mice in comparison to WT animals (Figure 4D). Similar to the results obtained for isoform 1, Ngln3 mRNA isoforms 2 was upregulated in all nine brain regions of *PWScr^m+/p−^* mice. The highest expression level of isoforms 2 was detected in the hippocampus, isocortex, and olfactory bulb as well as the thalamus and hypothalamus (Figure 4D). At this point, we cannot explain why the differences in the Nlgn3 mRNA isoforms 1 and 2 between mouse genotypes are more pronounced in comparison to the bulk mRNA. Perhaps there are other putative RNA isoforms that compensate the observed differences. In any event, the upregulation of both Nlgn3 mRNA isoforms in *PWScr^m+/p−^* mice raised reasonable doubts about a direct involvement of the Snord116 in the regulation of Nlgn3 posttranscriptional processing in vivo. In mice, Snord116 expression is restricted to the central nervous system (Figure 3E). Baldini et al. performed the Snord116 knock-down experiments in the human HeLa S3 cell line, originating from a cervical carcinoma [45]. Hence, we cannot exclude that the previously reported results are due to innate differences in the model systems.

## 3. Discussion

The contribution of an individual anatomical region to the total RNA amount isolated from the brain is roughly proportional to its size. Therefore, any region-specific regulation of expression, especially in relatively smaller regions (e.g., hypothalamus), may be masked and not identified. Most brain regions are functionally specialized. Differential regulation of gene expression in specific anatomical regions (and even in individual cells) may itself provide new mechanistic insights into the multiple biochemical pathways associated with complex brain functions. This is an important aspect to consider in the investigation of various neurological diseases, including PWS. Because of the cellular heterogenicity of the anatomical regions, leading to differences in gene expression profiles, and the numerous changes in the transcriptome caused by PWS, finding genes that are simultaneously stably expressed across various brain regions and at the same time stably expressed in any given region in both genotypes proved to be challenging. For example, while we found *Alg5* to be exceptionally stable in our analysis, it still ranked comparatively low in the midbrain when comparing expression stability between WT and *PWSc*r*^m+/p−^* samples (Figure 2F). Likewise, *Gusβ* is very stably expressed in almost all brain areas with the noteworthy exception of the olfactory bulb, where it is actually the least stable gene, according to our results (Figure 2F). Therefore, a sensible approach would be to tailor one’s selection of housekeeping genes to the particular brain region(s) examined. If only one or a few brain areas are of interest, an appropriate set of genes can easily be chosen by combining the ones with the lowest (i.e., best) stability ranking in those respective areas (Figure 2F and Appendix A).

Most PWS patients are deficient in GH, which has far-reaching consequences for the resulting symptoms, i.e., shorter stature, decreased muscle mass and increased fat mass, and lower muscle tone. [1,9,10]. Therefore, GH therapy is recommended and regularly prescribed as a treatment [8,53,54]. However, the mechanistic connection between the non-protein coding RNAs of the PWS locus, and the GH axis is still unclear. GH signaling itself is modulated by GH binding proteins, which compete with the GH downstream effector IGF-1 and block the respective IGF-1 receptor [55]. Levels of circulating IGFBP7 were reported to be increased 1.9-fold in PWS patients aged one to eight years without GH treatment and expression of *IGFBP7* was elevated in iPSC-derived neurons obtained from PWS patients [13]. Likewise, the serum Igfbp7 level was increased 2.2-fold, and *Igfbp7* gene expression was elevated in the brain, but not in the liver, heart, or adipose tissue of *PWScr^m+/p−^* mice compared to their wild-type siblings [13]. This is intriguing since coincidentally, Snord116 expression in mice is also *de facto* limited to the central nervous system (Figure 3E) (unlike in humans, where it is also expressed at various levels in other tissues) [2,7]. Our data somewhat confirm these results, as we found the mRNA level of Igfbp7 to be elevated in all analyzed brain regions in *PWScr^m+/p−^* mice compared to WT siblings on postnatal day 28, although statistical significance was reached only in the hippocampus, hypothalamus, and cerebellum (Figure 3A). Although the mechanism of Snord116-mediated regulation of *Igfbp7* remains unclear, or whether the observed upregulation has any functional significance, at least the correlation can be used to formulate a hypothetical connection between the non-protein coding RNAs of the PWS critical region and the GH axis. 

The activity of IGFBP7 is modulated by proteolytic cleavage. The N-terminally cleaved protein has a reduced affinity for the IGF-1 receptor and is unable to compete with IGF-1 and prevents GH signal cascade activation [55,56,57,58]. Cleavage can be facilitated by a number of proteases, among them members of a family of proconvertases PCSK1 (PC1) and PCSK2 (PC2). *PSCK1* expression is downregulated in iPSC-derived neurons from PWS patients and in the hypothalamus of fasting *Snord116^m+/p−^* mice [12]. In the *Snord116^m+/p−^* mouse model [59], this resulted in altered prohormone (proinsulin, pro-GH-releasing hormone and proghrelin) processing, mainly by insufficient processing of an inactive prohormone into its active form [12]. There are several documented cases of patients with inactivating *PSCK1* mutations that had symptoms very similar to the second phase of PWS in humans, i.e., obesity, hypogonadotropic hypogonadism and a dysregulated endocrine axis [60]. The results in mice are mixed, since homozygous *Pcsk1* knock-out mice do not develop obesity but dwarfism [61]. The growth retardation could be due to insufficient processing of pro-GH releasing hormone and a possible explanation for the lack of weight gain could be the low levels of circulating active insulin in the *Psck1*^−/−^ mice. Our comparative analysis of *Pcsk1* gene expression revealed its ~2-fold downregulation in the hypothalamus in *PWScr^m+/p−^* mice compared to wild-type siblings. This might correspond, at least in part, to the phenotype observed in *PWScr^m+/p−^* mice, which are growth retarded but do not develop obesity. Interestingly, the hypothalamic neuroendocrine dysfunction is considered a hallmark of PWS and is responsible for patient’s difficulties in controlling food intake and body weight management. The impaired *PCSK1* activity may indeed explain some of the symptoms observed in PWS [62]. However, unlike previous results investigating iPSCs from PWS patients, we found that *Pcsk2* gene expression is slightly upregulated in several brain regions of *PWScr^m+/p−^* mice. The consequences of increased Pcsk2 mRNA levels observed in the isocortex and, to a lesser extent, in the olfactory bulb, hippocampus, and medulla remain unclear and need to be examined further. This prohormone convertase is less well studied, and mutations in *PCSK2* are commonly associated with impaired glucose homeostasis in the context of type 2 diabetes [63,64,65]. The *PWScr^m+/p−^* mice have a mostly normal glucose metabolism, while human PWS patients are thought to suffer from type 2 diabetes, especially in the late stage of disease progression, typically due to severe obesity [6,66].

### Potential Interactions of Snord116 with Predicted mRNA Targets

The base complementarity of Snord116 to Nhlh2 mRNA postulating their molecular interaction has recently been reported [50]. Experiments using plasmid-based expression of Snord116 in the mouse hypothalamic cell line N29/2 showed that snoRNA co-expression increased Nhlh2 mRNA stability [50]. In the mouse brain regions analyzed, we did not observe such a stabilization effect. In the hypothalamus of *PWScr^m+/p−^* mice, Nhlh2 expression was not statistically different from their WT control (Figure 3D). Moreover, Nhlh2 transcript levels in two brain regions, the cerebellum and midbrain, were about twice as high in *PWSc*r*^m+/p−^* mice lacking Snord116 expression compared to wild-type siblings (Figure 3D). The physiological implications of these mRNA changes are unclear, and the functional relevance, if any, remains to be further investigated at the protein level. Given the uniform expression of Snord116 in various brain regions, all of the above may serve as an argument against the posttranscriptional stabilization of Nhlh2 mRNA by Snord116 in vivo (Figure 3D). Another important aspect of biological relevance that is often overlooked and rarely adequately reproduced in in vitro overexpression experiments is the proper subcellular localization of transcripts. C/D box snoRNAs, such as Snord116, are processed from their hosts’ transcript and readily transported to the nucleolus [2,67,68]. In contrast, mRNAs of protein-coding genes, such as Nhlh2, are directed for translation and thus exported from the nucleus almost immediately after processing. Under physiological conditions, it is doubtful that these two species of RNA will ever meet in amounts sufficient to account for any meaningful interaction, i.e., stabilization of mRNA that would influence the cellular levels of Nhlh2 protein. In addition, the *Nhlh2* gene is differentially expressed across the investigated brain regions and was barely detectable in the isocortex of P28 mice, whereas Snord116 is ubiquitously expressed in the murine central nervous system (Figure 3E). Although this in itself is not a strong argument against the interaction, it raises reasonable doubts about the hypothesis that Snord116 plays a significant role in regulating Nhlh2 mRNA stability, which would require that the activity of both genes be at least somewhat correlated in the corresponding affected tissues. While we cannot completely rule out a function of Snord116 in regulating *Nhlh2* gene expression at some stages of mouse development, it seems increasingly unlikely that any form of direct interaction would make a significant contribution to the pathogenesis of PWS. Notably, similarly to our results, a recent study also found no significant differences in the hypothalamic expression of *Nhlh2* gene between *Snord116^m+/p−^* and wild-type mice (Figure 3) [69,70].

Likewise, an evolutionary conserved complementarity of the Snord116 targeting region to the alternatively spliced exon 3 of Nlgn3 pre-mRNA was recently reported. Snord116 knock-down ex vivo experiments using the HeLa S3 cell line showed increased expression of the Nlgn3 exon 3 containing mRNA isoform [45]. Although we could also observe increased levels of murine Nlgn3 transcripts in all investigated brain regions of *PWScr^m+/p−^* mice, the effect was not specific to this particular isoform, but rather observed in all assayed isoforms, and thus, did not correlate with Snord116-guided regulation of exon 3 exclusion. Thus, our results raise reasonable doubt on the direct involvement of Snord116 in posttranscriptional regulation of Nlgn3. The consequences of increased Nlgn3 transcript remain unclear, but Nlgn3 is implicated in a number of behavioral impairments, such as autism spectrum disorders (ASD). One example is the Arg^451^→Cys^451^ (R451C) mutation in *NLGN3*, which was reported in an ASD patient [71]. In transgenic mice, the R451C mutation caused an increase in inhibitory synaptic transmission in the somatosensory cortex, which was accompanied by enhanced spatial learning abilities and disturbed social behavior [72]. This is likely a gain-of-function mutation since upon genetic deletion of *Nlgn3,* no comparable effect could be observed [72]. Interestingly, a similar imbalance of excitatory to inhibitory neuronal activity was also observed in *Magel2*-deficient mice [73]. In addition, increased spatial learning ability was also observed in one of the *Necdin* deficient mouse models [6,74]. Since no such behavioral alterations have been reported for *PWScr^m+/p−^* or *Snord116^m+/p−^* knock-out mouse models, it cannot be excluded whether perturbations in neuronal activity induced via a Snord116-Nlgn3 interaction may contribute to the complex behavioral symptoms observed in PWS patients [5,6,59].

Notably, *Snord116* gene knock-down using transfected antisense oligonucleotides reduced detectable Snord116 levels by about 50% in the HeLa cells. However, it was shown that an amount of only about 15% of Snord116 transcribed from the reactivated maternal allele was already sufficient to compensate the growth retardation phenotype in *PWScr^p^*^−*/m5′LoxP*^ mice [75]. Therefore, a meaningful experimental model system representing the pathophysiology of PWS would probably require a reduction in Snord116 levels below an amount which we know is already compensatory regarding at least the main symptoms.

Reports of the effect of Snord116 on the ratio of spliced mRNA isoforms are somewhat reminiscent of the Snord115 story, which began first with a hypothesis followed by cell culture experiments showing SNORD115 involvement in the regulation of alternative splicing of serotonin receptor 2C (5-Ht2cr) pre-mRNA [2,76]. Notably, the putative SNORD115–pre-mRNA regulation in cell culture experiments was observed only when the original splice site was mutated for optimal splicing [76]. SNORD115 is another imprinted orphan C/D box snoRNA located in the *PWScr* region whose expression is disrupted in typical PWS patients, i.e., long deletions and maternal UPD. The persistent hunger and the inability to feel satiated is one of the major medical problems that leads to morbid obesity in PWS patients. Regulation of appetite, satiety, and mood are pathways involving 5-Ht2cr signaling. Consequently, the involvement of Snord115 RNA in the regulation of 5-Ht2cr signaling has been suggested among the causes of PWS [76].

Although there were some doubts expressed earlier [77,78,79], the first actual counterevidence was discovered more than a decade later when RNA-seq data from postmortem hypothalamic tissue of PWS patients did not show any changes in alternative splicing of 5-HT2C receptor pre-mRNA [80]. Furthermore, no significant differences were found in the alternative splicing of 5-Ht2cr pre-mRNA when *Snord115* was ectopically expressed in mouse choroid plexus in vivo. [81]. The 5-Ht2cr alternative splicing hypothesis was finally laid to rest after the thorough analysis of a recently established Snord115 knock-out mouse model [82]. No changes in alternative splicing of 5-Ht2cr pre-mRNA or significant differences in energy balance or food uptake were found in mice, raising doubt on the causal involvement of Snord115 in major PWS symptoms [81,82].

To the best of our knowledge, to date, none of the previously predicted targets of orphan snoRNAs from the PWS locus [83] have been validated and confirmed as having clinical relevance in vivo. In particular, the biological role of orphan Snord116 remains elusive until now, even after the scientific community has spent considerable time and efforts analyzing it. This is somewhat remarkable because the disruption of the *SNORD116* gene cluster causes the major symptoms observed in PWS patients and pronounced growth retardation in mice. It has been more than two decades since the discovery of SNORD116 RNA and the identification of the PWS critical region (*PWScr*) as the main culprit in causing PWS. Finally, the generation of a compensatory mouse model has strongly suggested the causal significance of Snord116 in the pathogenesis of PWS, although the involvement of another non-protein coding transcript consisting of spliced Ipw-A exons (SNORD116 host transcript or SNHG14) cannot be excluded. Of the several strategies used in generating compensatory mouse models, so far, only the reactivation of a maternal allele has produced the desired effect of actually compensating for the underlying symptoms [6,75]. This further emphasizes the importance of genomic organization, tissue and/or cell-type specificity of expression with respect to RNA function under physiological conditions. Given enough scrutiny, a detailed analysis of the perturbed pathways should eventually reveal the mechanism through which the absence of *SNORD116* causes PWS.

Recent studies have focused on a putative interaction of Snord116 RNA with different mRNA targets [45,50]. The canonical function of most snoRNAs is to guide the posttranscriptional processing and modification of ribosomal RNAs; particularly, C/D-box snoRNAs are involved in 2′-O-ribose methylation. Interestingly, it has recently been shown that the canonical C/D-box snoRNA, SNORD13 can also mediate the acetylation of cytidine (ac4C1842) of 18S rRNA [84]. Moreover, this snoRNA function is evolutionarily conserved between yeast, fruit fly, zebrafish and human [85]. In addition to the predicted mRNA targets, Snord116 exhibits antisense-base complementarity to 18S rRNA [86]. However, analysis of rRNA ribose methylations did not reveal any major differences between mice harboring a long *Snord116-Snord115* genes cluster deletion and wild-type controls at embryonal day 16.5 [82]. On the other hand, 2′-O-ribose methylation (Nm) also occurs on mRNA affecting the posttranscriptional regulation of gene expression [87]. Interestingly, there are reports suggesting the involvement of C/D-box snoRNAs in complex with methyltransferase fibrillarin in Nm modification of mRNAs [87,88]. As of yet, the putative changes in transcriptome posttranscriptional nucleotide modification profiles of Snord116 knock-out mouse models and/or PWS patients have not yet been reported. Consequently, this could be a worthwhile approach to investigate the potential biochemical function of SNORD116 for understanding the molecular basis of Prader–Wi.

## 4. Materials and Methods

### 4.1. Animals and Tissue Collection

All mouse procedures were performed in compliance with the guidelines for the welfare of experimental animals issued by the Federal Government of Germany and approved by the State Agency for Nature, Environment and Consumer Protection North Rhine-Westphalia LANUV (Landesamt für Natur, Umwelt und Verbraucherschutz Nordrhein-Westfalen).

Animals were kept in specific pathogen-free conditions at the Transgenic animals and genetic engineering models (TRAM) core facility of the University Clinic Muenster in a temperature (21 °C) controlled room with 12:12 h light/dark cycle in standard sized (36 (l) × 20 (w) × 20 (h) cm), individually ventilated cages.

Wild-type C57BL/6 female mice were crossed with *PWScr^m+/p−^* male animals of the same genetic background. The resulting wild-type and *PWScr^m+/p−^* siblings were weaned between 19 and 23 days after birth and genotyped as described by Skryabin et al. [5]. Brain tissue was collected from male mice at postnatal day 28 (P28).

### 4.2. RNA Isolation and cDNA Synthesis

Total RNA was purified from tissue samples with the RNA Clean & Concentrator kit system (Zymo Research, Irvine, CA, USA, #R1017) using the TRIzol (TRI Reagent, Zymo Research, #R2050-1-200) extraction method according to the manufacturers’ instructions. Briefly, 800 µL of TRIzol reagent was added to a frozen sample, which was then homogenized, vortexed and incubated for 5 min at room temperature. Subsequently, 160 µL (0.2 × Vol of Trizol) of chloroform was added, and the mixture was vortexed thoroughly and spun for 10 min at 12,000× *g*. The resulting aqueous upper phase was removed to a new tube, mixed with the same volume of absolute ethanol and transferred to a column (Zymo-Spin IIC Column, Zymo Research, #C1011). The RNA was washed according to the protocol and eluted with 50 µL of ddH_2_O. The concentration was measured immediately with a NanoDrop spectrophotometer ND-1000 (Thermo Scientific, Waltham, MA, USA).

Synthesis of cDNA was performed with an aliquot of 2.5 µg total RNA using a random hexamer primer mix and M-MuLV reverse transcriptase (New England Biolabs, Ipswich, MA, USA, #M0253) at 52 °C for 1 h.

### 4.3. RT-qPCR

Quantitative PCR was performed on a LightCycler 480 system (Roche Diagnostics, Basel, Switzerland) using Luna Universal qPCR Master Mix (New England Biolabs, #M3003) and LNA-based, FAM labeled hydrolysis probes (Universal ProbeLibrary, Roche Diagnostics). Assay details, including primer and probe sequences, are listed in Appendix A. Measurements were performed with 1 µL of a 1:5 cDNA dilution of at least three biological samples from each genotype in technical triplicate. Crossing point (Cq) values were automatically calculated using the LightCycler 480 SW 1.5 software (Roche). Gene expression is represented as mean ± SD, relative to the average of the housekeeping panel consisting of Alg5-Hmbs-Gusβ (2^−ΔCq^) (Appendix A).

### 4.4. Ranking of Expression Stability and Statistical Analysis

Expression stability was assessed using the comprehensive inline ranking package RefFinder [49], which is based on a combination of the four methods geNorm [38], Normfinder [40], BestKeeper [39] and comparative delta Ct [37]. Briefly, each of the four methods was used to assign a stability rank to each of the eight candidate genes ranging from 1 (most stable) to 8 (least stable). The final ranking is then obtained by calculating the geometric mean of the stability ranks for each gene. The whole process was repeated for each condition (i.e., expression stability across all nine brain regions within each of the two genotypes and expression stability across the two genotypes within each of the nine regions).

Statistical analysis was done in Excel (Excel for Mac 2011 v14.6.8, Microsoft, Redmond, WA, USA) and RStudio (RStudio v1.4.1106, PBC, Boston, MA, USA). Normality of the data was verified using the Shapiro–Wilk method, and the statistical significance of the differences between genotypes was tested with Student’s *t*-test.

### 4.5. Northern Blot Analysis

Northern blot analysis was performed as described previously [81]. Briefly, total RNA was purified using Trizol reagent; 2 µg of total RNA was separated on a denaturing polyacrylamide gel (8% PAA, 7 M Urea) and transferred onto a nylon membrane (BrightStar Plus, Southampton, UK; Ambion, Austin, TX, USA, #AM10100). Oligonucleotide probes (Appendix A) were labeled with γ-^32^P-ATP using T4 polynucleotide kinase (New England Biolabs, #M0201). Membranes were pre-hybridization in 0.5 M sodium phosphate buffer containing 7% SDS (pH 6.5) at 56 °C for 30 min. Hybridization was performed with 50 pmol ^32^P labeled specific probes in a pre-hybridization buffer for overnight at 56 °C. Subsequently, membranes were washed three times for 30 min at 46 °C with 0.1 M sodium phosphate containing 1% SDS and exposed to MS film overnight at −80 °C.

## Figures and Tables

**Figure 1 ijms-23-08729-f001:**
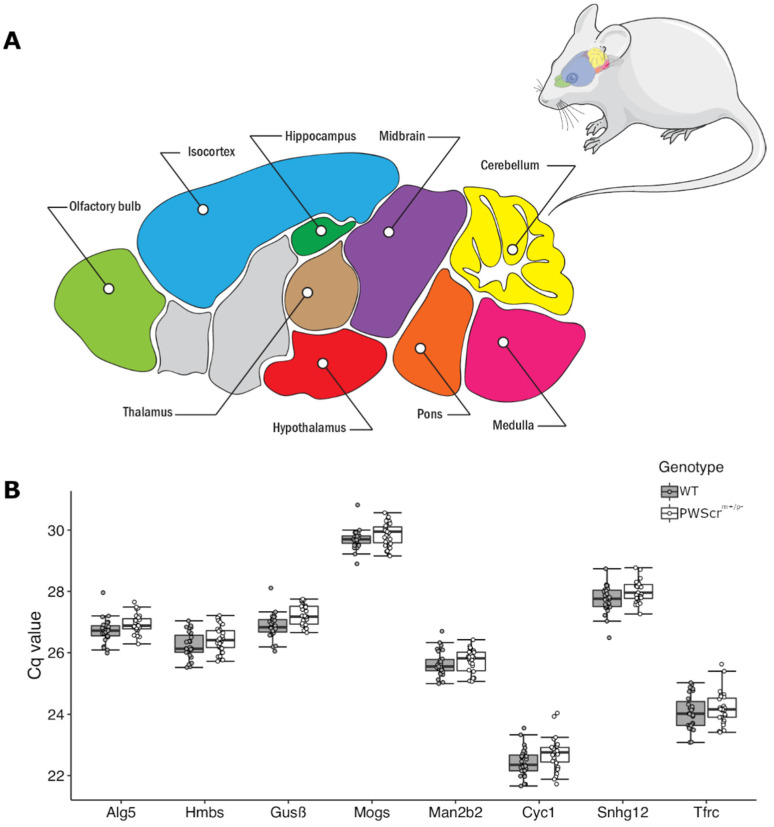
**Mouse brain regions and expression of selected reference gene candidates.** (**A**) Schematic representation of a sagittal section of an adult mouse brain with nine analyzed regions (colored and labelled). Areas that were not analyzed are shown in gray. (**B**) The resulting Cq values of the selected eight reference gene candidates in all nine brain regions analyzed (mean ± SD). Wild-type bars are depicted in dark gray, *PWScr^m+/p—^* in white.

**Figure 2 ijms-23-08729-f002:**
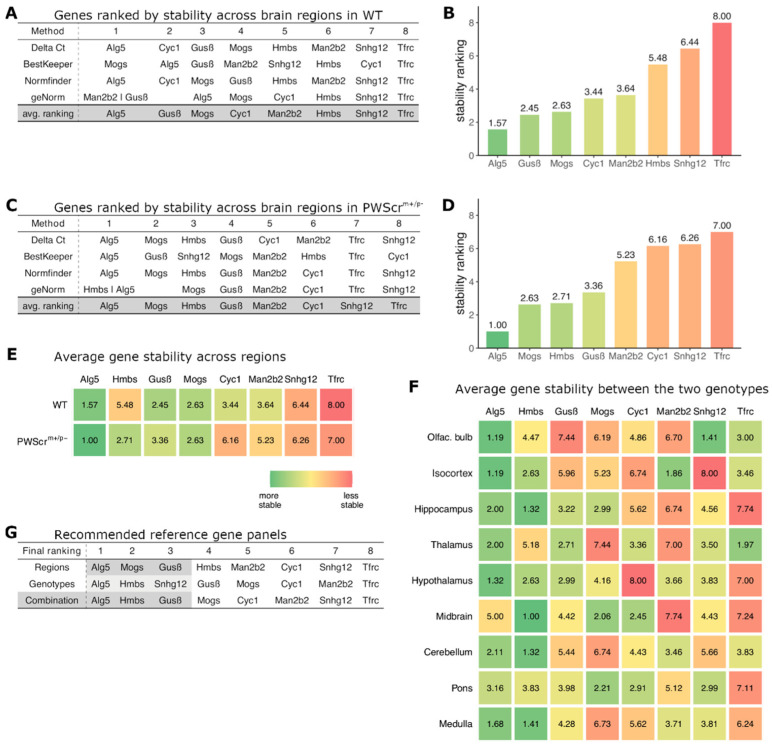
**Identification of the reference gene panel.** (**A**,**C**) Expression stability ranking across nine brain regions of reference gene candidates in WT and *PWScr**^m+/p−^* mice, respectively, determined by four different stability ranking methods. Avg. ranking: shows the calculation of the resulting average ranking of the candidate genes (highlighted in gray; bottom row). (**B**,**D**) Average ranking position of each of the genes (lower average rank/green is more stable than higher average rank/red) across the nine brain regions of WT (**B**) and *PWScr**^m+/p−^* (**D**) mice, respectively. (**E**) Average expression stability ranking positions of gene candidates across the nine brain regions in WT (top) and *PWScr^m+/p^*^−^ (bottom) mice. (**F**) Average ranking of expression stability between the two genotypes (WT and *PWScr^m+/p^*^−^) within each of the nine brain regions for the reference gene candidates. (**G**) Compilation of reference gene panel. Regions: average ranking of genes for expression stability within regions of genotypes; genotypes: average ranking of expression stability across investigated genotypes within each brain region; combination: recommended panel of reference genes, i.e., the three highest ranked genes, highlighted in gray.

**Figure 3 ijms-23-08729-f003:**
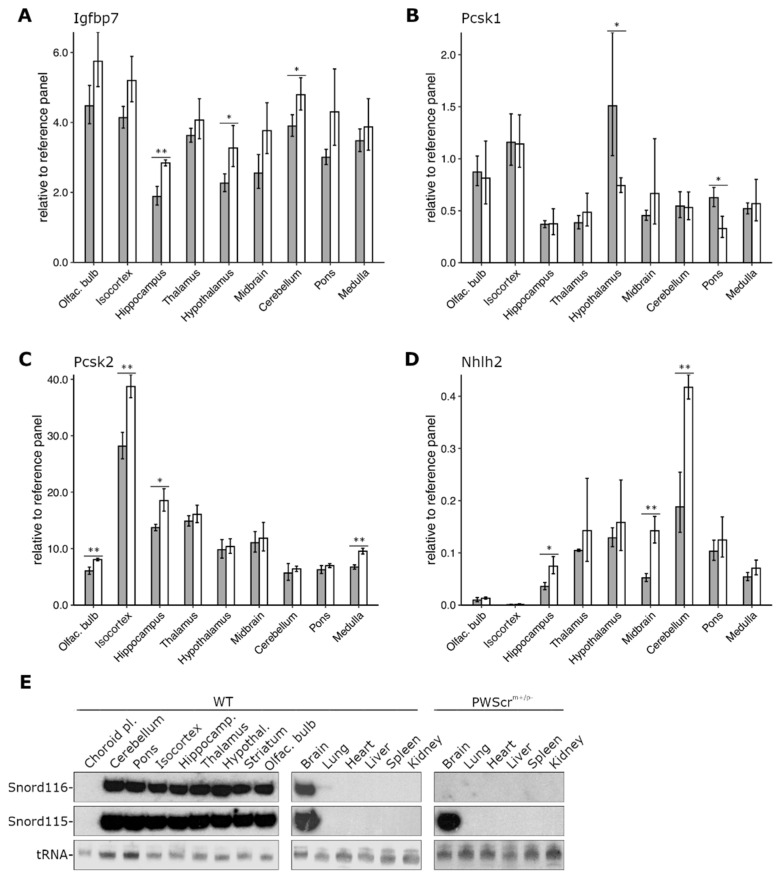
**Expression analysis of *Pcsk1*, *Pcsk2*, *Igfbp7* and *Nhlh2*.** (**A**) Comparative expression analysis of *Igfbp7*, (**B**) *Pcsk1*, (**C**) *Pcsk2*, and (**D**) *Nhlh2* genes relative to reference gene panel in nine brain regions of male mice P28 (WT in gray, *PWScr^m+/p−^* in white). Mean ± SD of three biological replicates (six for WT hypothalamus); * *p* ≤ 0.05, ** *p* ≤ 0.01, unpaired two-samples *t*-test. (**E**) Northern blot hybridization analysis of Snord116 and Snord115 expression in brain areas and tissues of WT mice (left and middle panels, respectively); and *PWScr^m+/p−^* (right panel) animals. As a loading control, tRNAs (negative image of an ethidium bromide-stained gel) is shown at the bottom.

**Figure 4 ijms-23-08729-f004:**
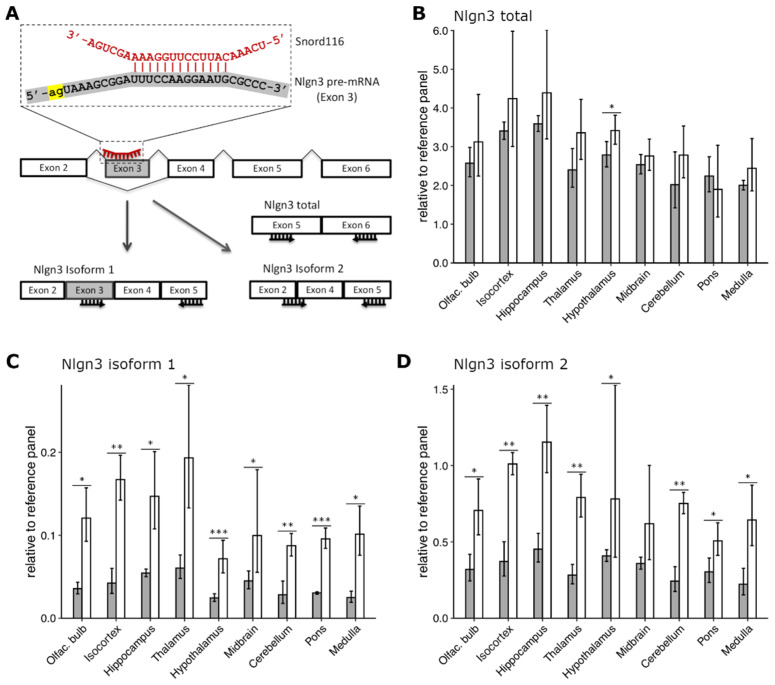
**Effect of Snord116 knock-out on expression and alternative splicing of Nlgn3 in mouse brain.** (**A**) Putative complementarity of Snord116 (red) with exon 3 of Nlgn3 pre-mRNA (gray). 3′-splicing site (acceptor) indicated in yellow. Schematic representation of alternative splicing of Nlgn3 mRNA into isoform 1 (including exon 3) and isoform(s) 2 (excluding exon 3), respectively. Forward and reverse primers for the designed RT-qPCR assays are depicted as black arrows under the exons. (**B**) Expression levels of total Nlgn3, (**C**) Nlgn3 isoform 1, and (**D**) Nlgn3 isoform 2 relative to the reference gene panel in nine brain regions of WT and *PWScr^m+/p−^* mice are depicted with gray and white bars, respectively, as Mean ± SD of three biological replicates (six for WT hypothalamus); * *p* ≤ 0.05, ** *p* ≤ 0.01, *** *p* ≤ 0.001, unpaired two-samples *t*-test.

## Data Availability

Not applicable.

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
