# Peer review of "Reference Genes across Nine Brain Areas of Wild Type and Prader-Willi Syndrome Mice: Assessing Differences in Igfbp7, Pcsk1, Nhlh2 and Nlgn3 Expression"

_ijms, 2022, doi:10.3390/ijms23158729_

Round 1
Reviewer 1 Report
Using four independent methods (Delta Ct, Best-23, Keeper, Normfinder and Genorm), the authors searched for housekeeping genes (n=11) which could be used as robust reference genes for RT-qPCR analyses in the mouse brain, including manually-dissected brain regions (n=9). Using SNORD116-KO mice (PWScr mice), they then applied their new set of identified reference genes to measure expression levels of a few genes (Igfbp7, Nlgn3, Pcsk1, Pcsk2, Nhlh2) predicted to be regulated – directly or indirectly - by a brain-specific box C/D snoRNA, namely SNORD116. Contrary to previous findings, their data do not support a direct role of SNORD116 in targeting Nlgn3 and Nhlh2 for post-transcriptional regulation in vivo. I believe that such observations are useful for the scientific community interested in the biology of SNORDs and/or the etiology of Prader-Willi syndrome. Indeed, I really appreciated the fact that the authors performed their analyses with biologically relevant tissue. This work is essential because it gives indications of "what SNORD116 does" in vivo, whereas previous published studies conducted in vitro with mostly non-neuronal cell systems indicate what “SNORD116 can do" which, in my opinion, constitutes a proof of principle but certainly not physiological arguments. More generally, the identification of robust housekeeping genes could also be beneficial for other studies dealing with gene expression in the brain.
I have no major comments to make. For ease of reading, however, I can offer the following minor recommendations.
Page 3, lane 110 – “However, even at the onset of the qPCR era, it was realized that these genes were often poor choices for the normalization of expression data [33–36]”. I would recommend that the authors be a little more explicit and explain what specific observations indicate that these genes, very often used by the scientific community, are not good choices.
Page 4, lane 189 - “section may be divided by subheadings. It should provide a concise and precise description of the experimental results, their interpretation, as well as the experimental conclusions that can be drawn”. What does it mean? I guess it was to test my …alertness. :+)
As the authors rightly point out, the changes in gene expression between WT vs SNORD116-KO are relatively small. If possible, I suggest that the authors look at the protein levels in the tissues for which the observed changes are the most important, i.e. Pcsk1 in hypothalamus and/or Nhlh2 in cerebellum (optional). If this is not possible, I strongly recommend that the authors explicitly state that the functional relevance, if any, of the observed changes at the mRNA level remains to be further demonstrated at the protein level.
Page 9, lane 320 - “Recent cell culture experiments reported that NHLH2 mRNA is a potential target for SNORD116 RNA [12,50]”. I suggest that the authors mention the cellular model used in the cited studies. If I understand, these are not neuronal models.
Page 9 (title) - “Snord116 influences abundance but not splicing of Nlgn3 mRNA in vivo”. I suggest to be more factual and write something like “lack of Snord116 influences abundance but not splicing of Nlgn3 mRNA in vivo”
Discussion, lane 437 - “Our data somewhat confirm these results, as we found transcription of Igfbp7 to be elevated in all analyzed brain regions in PWScrm+/p- mice compared to WT”. As I understand it, the authors noted an increase in the abundance of Igfbp7 mRNA. This does not necessarily mean that the transcription rate of Igfb7 gene is increased. This sentence needs to be changed.
Discussion section, lanes 445-475 - It seems to me that a summary diagram of the putative functional links between GH axis, SNORD116, Psck1/psck2, Igfbp7 (proteolytic cleavage by Psck1/2), and Nhlh2 (upstream regulator of Nhlh2) would be useful for a non-expert reader. This could also possibly include Nlgn3. Such a diagram could also summarize the changes (or not) observed between KO and WT mice.
Discussion, lane 449 - “PSCK1 expression is downregulated in iPSC-derived neurons from PWS patients and in the hypothalamus of fasting Snord116m+/p- mice [12]”. Is there any experimental evidence showing that the genes studied in this study are deregulated in PWS patients? The authors could have a look this publication: "A Transcriptomic Signature of the Hypothalamic Response to Fasting and BDNF Deficiency in Prader-Willi Syndrome. "Cell Rep. 2018 Mar 27;22(13):3401-3408. doi: 10.1016/j.celrep.2018.03.018.
Discussion, lane 514 - “Although we could also observe increased levels of murine Nlgn3 transcripts in all investigated brain regions of PWScrm+/p- mice, the effect was not specific to this particular isoform, but rather observed in all assayed isoforms, and thus, did not correlate with Snord116 regulation of exon 3 exclusion”. Do the authors believe that SNORD116 regulates directly the stability of Nlgn3 transcripts without affecting alternative splicing? In my view, the authors should be more explicit: is Nlgn3 a credible direct mRNA target for SNORD116?
Discussion, lane 588 - “In addition to the predicted mRNA targets, Snord116 exhibits antisense-base complementarity to 18S rRNA [86], however, the putative changes in transcriptome posttranscriptional base modification profiles of Snord116 KO mouse models and/or PWS patients, have not yet been reported”. As I understand it, the authors in Ref#86 predicted that SNORD116 directs ribose methylation on rRNA, not base modification. On the other hand, this putative rRNA site is not known to be 2'-O-methylated in human, at least based on recent RiboMeth-seq analyses. Finally, the authors of Ref#82 analyzed the profile of ribose methylations of rRNA in the brain of a double KO mouse strain deleted for both SNORD116 and SNORD115 genes. No major differences could be revealed between the two genotypes. This information should be indicated.
Author Response
We wish to thank the referees for their encouraging comments and constructive suggestions. Please find our point-by-point response (in red) below. Following the referees’ suggestions, we have introduced changes in the main text (all changes are marked in red).
Referee 1:
Using four independent methods (Delta Ct, Best-23, Keeper, Normfinder and Genorm), the authors searched for housekeeping genes (n=11) which could be used as robust reference genes for RT-qPCR analyses in the mouse brain, including manually-dissected brain regions (n=9). Using SNORD116-KO mice (PWScr mice), they then applied their new set of identified reference genes to measure expression levels of a few genes (Igfbp7, Nlgn3, Pcsk1, Pcsk2, Nhlh2) predicted to be regulated – directly or indirectly - by a brain-specific box C/D snoRNA, namely SNORD116. Contrary to previous findings, their data do not support a direct role of SNORD116 in targeting Nlgn3 and Nhlh2 for post-transcriptional regulation in vivo. I believe that such observations are useful for the scientific community interested in the biology of SNORDs and/or the etiology of Prader-Willi syndrome. Indeed, I really appreciated the fact that the authors performed their analyses with biologically relevant tissue. This work is essential because it gives indications of "what SNORD116 does" in vivo, whereas previous published studies conducted in vitro with mostly non-neuronal cell systems indicate what “SNORD116 can do" which, in my opinion, constitutes a proof of principle but certainly not physiological arguments. More generally, the identification of robust housekeeping genes could also be beneficial for other studies dealing with gene expression in the brain.
I have no major comments to make. For ease of reading, however, I can offer the following minor recommendations.
Page 3, lane 110 – “However, even at the onset of the qPCR era, it was realized that these genes were often poor choices for the normalization of expression data [33–36]”. I would recommend that the authors be a little more explicit and explain what specific observations indicate that these genes, very often used by the scientific community, are not good choices.
For clarification of this point we have modified the text accordingly: “However, even at the onset of the qPCR era, it was realized that these genes were often poor choices for the normalization of expression data since their mRNA levels vary by as much as 15-fold between tissues (i.e. GAPDH), are highly influenced by common cell culture components (i.e. ACTB) and also could be affected by any dysregulation of RNA polymerases I (i.e. rRNAs) and II, as is the case in numerous cancers [33–36]”.
Page 4, lane 189 - “section may be divided by subheadings. It should provide a concise and precise description of the experimental results, their interpretation, as well as the experimental conclusions that can be drawn”. What does it mean? I guess it was to test my …alertness. :+)
By an unfortunate oversight, after inserting the manuscript text into the ijms template, this template text remained. The text fragment was deleted.
As the authors rightly point out, the changes in gene expression between WT vs SNORD116-KO are relatively small. If possible, I suggest that the authors look at the protein levels in the tissues for which the observed changes are the most important, i.e. Pcsk1 in hypothalamus and/or Nhlh2 in cerebellum (optional). If this is not possible, I strongly recommend that the authors explicitly state that the functional relevance, if any, of the observed changes at the mRNA level remains to be further demonstrated at the protein level.
We have indicated that: “In both PWScrm+/p- and WT mice, measured Pcsk1 levels were highest in hypothalamus, olfactory bulb and isocortex and lowest in hippocampus; however, the putative functional significance of the observed mRNA-level changes must be further demonstrated at the protein level”.
The notable lack of any protein-level analysis for Nhlh2 has been added: “Although the physiological implications of this are unclear, and the functional relevance, if any, of the observed changes at the mRNA level remains to be further demonstrated at the protein level”.
Page 9, lane 320 - “Recent cell culture experiments reported that NHLH2 mRNA is a potential target for SNORD116 RNA [12,50]”. I suggest that the authors mention the cellular model used in the cited studies. If I understand, these are not neuronal models.
Following the referee’s suggestion we have modified the sentence accordingly: “Recent cell culture experiments using iPSC derived neurons from PWS patients and the mouse hypothalamic N29/2 cell line reported that NHLH2 mRNA is a potential target for SNORD116 RNA [12,50].”
Page 9 (title) - “Snord116 influences abundance but not splicing of Nlgn3 mRNA in vivo”. I suggest to be more factual and write something like “lack of Snord116 influences abundance but not splicing of Nlgn3 mRNA in vivo”
We appreciate this valuable suggestion and changed the text accordingly: “Lack of Snord116 influences abundance but not splicing of Nlgn3 mRNA in vivo”
Discussion, lane 437 - “Our data somewhat confirm these results, as we found transcription of Igfbp7 to be elevated in all analyzed brain regions in PWScrm+/p- mice compared to WT”. As I understand it, the authors noted an increase in the abundance of Igfbp7 mRNA. This does not necessarily mean that the transcription rate of Igfb7 gene is increased. This sentence needs to be changed.
Completely agree with the referee and changed the sentence accordingly: “…as we found the mRNA level of Igfbp7 to be elevated…”
Discussion section, lanes 445-475 - It seems to me that a summary diagram of the putative functional links between GH axis, SNORD116, Psck1/psck2, Igfbp7 (proteolytic cleavage by Psck1/2), and Nhlh2 (upstream regulator of Nhlh2) would be useful for a non-expert reader. This could also possibly include Nlgn3. Such a diagram could also summarize the changes (or not) observed between KO and WT mice.
As the referee rightfully mentioned above that explicit conclusions should be drawn only after the expression of the corresponding protein products are examined, we would not feel comfortable to include a diagram at this point. A diagram would have more impact in conveying (here, an interim) finding, which still could be incomplete. The text we provided is more subtle in that respect.
Discussion, lane 449 - “PSCK1 expression is downregulated in iPSC-derived neurons from PWS patients and in the hypothalamus of fasting Snord116m+/p- mice [12]”. Is there any experimental evidence showing that the genes studied in this study are deregulated in PWS patients? The authors could have a look this publication: "A Transcriptomic Signature of the Hypothalamic Response to Fasting and BDNF Deficiency in Prader-Willi Syndrome. "Cell Rep. 2018 Mar 27;22(13):3401-3408. doi: 10.1016/j.celrep.2018.03.018.
We could not find any of the discussed genes of interest in the dataset of Bochukova et al.
Discussion, lane 514 - “Although we could also observe increased levels of murine Nlgn3 transcripts in all investigated brain regions of PWScrm+/p- mice, the effect was not specific to this particular isoform, but rather observed in all assayed isoforms, and thus, did not correlate with Snord116 regulation of exon 3 exclusion”. Do the authors believe that SNORD116 regulates directly the stability of Nlgn3 transcripts without affecting alternative splicing? In my view, the authors should be more explicit: is Nlgn3 a credible direct mRNA target for SNORD116?
We have modified the text accordingly: “Although we could also observe increased levels of murine Nlgn3 transcripts in all investigated brain regions of PWScrm+/p- mice, the effect was not specific to this particular isoform, but rather observed in all assayed isoforms, and thus, did not correlate with Snord116 - guided regulation of exon 3 exclusion. Thus, obtained results raise a reasonable doubt on direct involvement of the Snord116 in posttranscriptional regulation of Nlgn3.”
Discussion, lane 588 - “In addition to the predicted mRNA targets, Snord116 exhibits antisense-base complementarity to 18S rRNA [86], however, the putative changes in transcriptome posttranscriptional base modification profiles of Snord116 KO mouse models and/or PWS patients, have not yet been reported”. As I understand it, the authors in Ref#86 predicted that SNORD116 directs ribose methylation on rRNA, not base modification. On the other hand, this putative rRNA site is not known to be 2'-O-methylated in human, at least based on recent RiboMeth-seq analyses. Finally, the authors of Ref#82 analyzed the profile of ribose methylations of rRNA in the brain of a double KO mouse strain deleted for both SNORD116 and SNORD115 genes. No major differences could be revealed between the two genotypes. This information should be indicated.
We have modified this part of the text accordingly:
“In addition to the predicted mRNA targets, Snord116 exhibits antisense-base complementarity to 18S rRNA [86], however, analysis of rRNA ribose methylations did not reveal any major differences between mice harbouring a long Snord116-Snord115 genes cluster deletion and wild type controls at embryonal day 16.5 [82]. On the other hand, 2′-O-ribose methylation (Nm) also occurs on mRNA affecting the posttranscriptional regulation of gene expression [87]. Interestingly, there are reports suggesting the involvement of C/D-box snoRNAs in complex with methyltransferase fibrillarin in Nm modification of mRNAs [87,88]. As yet, the putative changes in transcriptome posttranscriptional nucleotide modification profiles of Snord116 KO mouse models and/or PWS patients have not yet been reported. Consequently, this could be a worthwhile approach to investigate the potential biochemical function of SNORD116 for understanding the molecular basis of Prader-Willi Syndrome.”
Reviewer 2 Report
The manuscript provides a very detailed investigation into gene expression and splicing of key genes, previously suggested to be involved in the pathophysiology of Prader-Willi syndrome, using appropriate mouse model. The study looks into comprehensive list of distinct brain areas, devises a very thorough control genes and methods selection, and as such addresses important questions for the PWS research field. The study design and results are very clear and convincing, the discussion is sufficiently detailed and measured interpretation of the results is presented, considering the complexity of the research question.
Minor edits: the Authors should check spelling (e.g. symptomatology) and some redundant text ( e.g.lines 189-191).
Author Response
We wish to thank the referee for their encouraging comments.
Referee 2:
The manuscript provides a very detailed investigation into gene expression and splicing of key genes, previously suggested to be involved in the pathophysiology of Prader-Willi syndrome, using appropriate mouse model. The study looks into comprehensive list of distinct brain areas, devises a very thorough control genes and methods selection, and as such addresses important questions for the PWS research field. The study design and results are very clear and convincing, the discussion is sufficiently detailed and measured interpretation of the results is presented, considering the complexity of the research question.
Minor edits: the Authors should check spelling (e.g. symptomatology) and some redundant text ( e.g.lines 189-191).
The text has been edited as suggested.